# Bazedoxifene, a Selective Estrogen Receptor Modulator, Promotes Functional Recovery in a Spinal Cord Injury Rat Model

**DOI:** 10.3390/ijms222011012

**Published:** 2021-10-12

**Authors:** Yiyoung Kim, Eun Ji Roh, Hari Prasad Joshi, Hae Eun Shin, Hyemin Choi, Su Yeon Kwon, Seil Sohn, Inbo Han

**Affiliations:** 1School of Medicine, CHA University, CHA Bundang Medical Center, Seongnam-si 13496, Gyeonggi-do, Korea; irenekim1102@gmail.com; 2Department of Neurosurgery, CHA University School of Medicine, CHA Bundang Medical Center, Seongnam-si 13496, Gyeonggi-do, Korea; morolro@naver.com (E.J.R.); tlsgodms223@naver.com (H.E.S.); littlechoi88@gmail.com (H.C.); syunkwon@naver.com (S.Y.K.); sisohn@cha.ac.kr (S.S.); 3Department of Physiology and Pathophysiology, Spinal Cord Research Centre, Rady Faculty of Health Sciences, University of Manitoba, Winnipeg, MB R3E 0W2, Canada; hariprasadjoshi10@gmail.com

**Keywords:** spinal cord injury, secondary injury, inflammatory response, remyelination, bazedoxifene acetate

## Abstract

In research on various central nervous system injuries, bazedoxifene acetate (BZA) has shown two main effects: neuroprotection by suppressing the inflammatory response and remyelination by enhancing oligodendrocyte precursor cell differentiation and oligodendrocyte proliferation. We examined the effects of BZA in a rat spinal cord injury (SCI) model. Anti-inflammatory and anti-apoptotic effects were investigated in RAW 264.7 cells, and blood-spinal cord barrier (BSCB) permeability and angiogenesis were evaluated in a human brain endothelial cell line (hCMEC/D3). In vivo experiments were carried out on female Sprague Dawley rats subjected to moderate static compression SCI. The rats were intraperitoneally injected with either vehicle or BZA (1mg/kg pre-SCI and 3 mg/kg for 7 days post-SCI) daily. BZA decreased the lipopolysaccharide-induced production of proinflammatory cytokines and nitric oxide in RAW 264.7 cells and preserved BSCB disruption in hCMEC/D3 cells. In the rats, BZA reduced caspase-3 activity at 1 day post-injury (dpi) and suppressed phosphorylation of MAPK (p38 and ERK) at dpi 2, hence reducing the expression of IL-6, a proinflammatory cytokine. BZA also led to remyelination at dpi 20. BZA contributed to improvements in locomotor recovery after compressive SCI. This evidence suggests that BZA may have therapeutic potential to promote neuroprotection, remyelination, and functional outcomes following SCI.

## 1. Introduction

Spinal cord injury (SCI) is a life-shattering condition that leads to severe and permanent neurological deficits. Numerous therapeutic approaches have shown potential in preclinical studies, but only a few have made progress due to the complexity of SCI pathophysiology [1,2,3,4,5]. It is well established that SCI follows a biphasic time course; the initial mechanical injury is caused by impact accompanied by persistent compression of the spinal cord, while the secondary phase is characterized by destructive and self-propagating biochemical changes in neuronal and glial cells that lead to increased dysfunction and cell death [6]. Among secondary injury cascades, demyelination is a major pathological feature of acute traumatic SCI. Demyelination is mainly a consequence of oligodendrocyte (OL) cell death and damage caused by multiple mechanisms: mechanical injury followed by toxic blood components, ischemic damage, oxidative stress, excitotoxicity, pro-inflammatory cytokines, and autophagy [7,8]. Demyelination is prevalent during the first 2 weeks post-injury and continues to impair the amplitude and speed of electrical conductance of injured axons [9,10,11,12]. Thus, progressive demyelination leaves axons vulnerable to degeneration, which could diminish the functionality of spared circuits and limit functional improvements [13]. Accordingly, remyelination appears to be a plausible strategy to protect spared axons from secondary damage and promote functional recovery [7].

Myelin regeneration occurs spontaneously beginning 2 weeks after injury, through a process in which surviving oligodendrocyte progenitor cells (OPCs) proliferate and differentiate to remyelinate, and this may continue for up to 3 months [14,15]. However, endogenous remyelination is often suboptimal and incomplete due to the following changes in the post-injury environment: (i) limited replacement of myelinating OLs by OPCs, (ii) insufficient levels of key growth factors for OL maturation, (iii) inadequate clearance of myelin debris, and (iv) inhibitory factors mainly driven by activated glia [8]. Despite vigorous attempts to discover therapies to enhance remyelination, the clinical relevance of remyelination and its contributions to functional recovery after SCI remain contentious. Nonetheless, the processes involved in remyelination are the mechanistic basis of several ongoing trials, and remyelination has become an important therapeutic target [8,16].

BZA, a third-generation selective estrogen receptor modulator (SERM), is an indole-based estrogen receptor ligand currently used to prevent and treat postmenopausal osteoporosis [17]. Recently, BZA has been identified as a compelling remyelinating agent with a highly tolerable safety profile in a multiple sclerosis model [18]. Although SERMs have previously been implicated in remyelination and neuroprotection by targeting nuclear estrogen receptors (ERs), BZA has been found to promote remyelination independently of ERs [18]. Furthermore, research has revealed multiple neuroprotective effects of BZA in various central nervous system (CNS) injury models. Specifically, BZA has been shown to reduce ischemic brain damage and apoptosis by modulating the mitogen-activated protein kinase (MAPK)/extracellular signal-regulated kinase (ERK)1/2 and phosphatidylinositol 3-kinase (PI3K)/protein kinase B (AKT) signaling pathways [19,20]. Moreover, BZA has been found to attenuate impaired cognitive function and increased blood-brain barrier (BBB) permeability by modulating inflammation via MAPK pathway suppression in a traumatic brain injury (TBI) model [21]. In fact, pro-inflammatory microglia and the MAPK/ERK pathway are closely related to demyelination and remyelination processes in the injured CNS, undermining OPC survival and differentiation [22,23,24].

Similarly, evidence suggests that the main human estrogen, 17β-estradiol (E2) is a significant multi-active neuroprotective agent that may reduce inflammation and apoptosis, promote angiogenesis and myelination, and improve motor function after SCI [25,26,27]. Yet, investigations regarding the effects of BZA in SCI are surprisingly lacking. Hence, the research exploring the therapeutic efficacy of BZA in animal models of CNS injuries spurred our interest in evaluating its possible neuroprotective and remyelinating effects in SCI. We hypothesized that BZA improves functional recovery from SCI by promoting remyelination via modulation of the MAPK pathway and inflammatory response. In this study, we revealed the therapeutic potential of BZA by investigating its impact on inflammation, angiogenesis, apoptosis, remyelination, and locomotor outcomes in vitro and in a rat model of SCI.

## 2. Results

### 2.1. BZA Promotes Survival of PC12 Neural Cells and Reduces In Vitro Nitric Oxide (NO) Production

To determine non-cytotoxic working concentrations of BZA, cell viability was measured by CCK-8 assay in PC12 cells. Compared to control and dimethyl sulfoxide (DMSO) group, lipopolysaccharide (LPS, 5 μg/mL) treatment significantly decreased cell viability from 100% to 44.03 ± 5.5%, but BZA 10 μM and BZA 20 μM treatments increased cell viability up to 74.25 ± 16.39% and 81.99 ± 23.1%, respectively (Figure 1A). However, the effect of BZA concentration was insignificant. The result indicates that both 10 μM and 20 μM BZA concentrations are optimal treatment concentrations for in vitro experiments.

The anti-inflammatory effect of BZA was evaluated in LPS-stimulated RAW 264.7 macrophage cells by a nitric oxide (NO) assay. The cells were treated with LPS and LPS + BZA (10 μM and 20 μM) and were examined 24 h after treatment. NO production increased steadily for 24 h after LPS treatment. The differences in NO production among the LPS-stimulated (non-treated), LPS + BZA 10 μM, and LPS + BZA 20 μM groups are presented in Figure 1B–D. As shown, NO production significantly decreased in the BZA-treated group in a dose-dependent manner compared with LPS-treated macrophage.

### 2.2. BZA Promotes In Vitro Inflammatory Resolution

The inflammatory response is a major pathologic feature of the secondary injury cascade after SCI. To evaluate the effect of BZA on the inflammatory response, the expression of pro-inflammatory (IL-6, TNF-α, CCL-2) and anti-inflammatory (IL-10) cytokines was measured in LPS-treated RAW 264.7 macrophages. As shown in Figure 2A–D, the expression levels of IL-6 (Figure 2A), TNF-α (Figure 2B), and CCL-2 (Figure 2C) were increased and that of IL-10 (Figure 2D) was decreased by LPS-induced inflammation. The increased levels of these pro-inflammatory cytokines driven by LPS were significantly impeded by BZA treatment, regardless of its concentration. However, the level of IL-10 significantly increased in the BZA-treated groups in a dose-dependent manner.

### 2.3. BZA Enhances Angiogenesis and Reduces In Vitro Blood-Spinal Cord Barrier (BSCB) Disruption

It is well established that blood–spinal cord barrier (BSCB) disruption after SCI allows blood cells to infiltrate the damaged parenchyma and exacerbates secondary injuries, such as focal edema, ischemia, focal hemorrhage, and inflammation [28]. Hence, maintaining BSCB permeability would induce a protective effect against secondary damage following SCI. The endothelial cells are connected by adhesion proteins and sealed by tight junction (TJ) proteins, which play an important role in maintaining the BSCB integrity [29]. In this study, we examined the changes of the TJ proteins including zonula occludens-1 (ZO-1) and occludin in a human cerebral microvascular endothelial cell line (hCMEC/D3) upon LPS-induced inflammation and evaluated the effects of BZA on these proteins with immunofluorescence imaging. As shown in Figure 3, LPS significantly decreased the levels of ZO-1 and occludin compared to the control. Subsequently BZA treatment significantly alleviated the reduction of ZO-1 and occludin expression. Similarly, disruption of the spinal cord vasculature is another factor that aggravates secondary injuries and reduces BSCB permeability after SCI [17]. Therefore, attempts to regulate angiogenic response and vascular stability have been made to promote neural regeneration and recovery [30,31,32]. To evaluate the effect of BZA on angiogenesis and vascular maintenance, the expression levels of angiopoietin-1 (ANGPT-1), von Willebrand factor (vWF), and α-smooth muscle actin (α-SMA) were determined in hCMEC/D3 cells. ANGPT-1 is crucial in limiting vascular permeability and controlling BSCB integrity, and eventually diminishing the inflammatory response by securing paracellular junctions [17]. Moreover, α-SMA is expressed in capillary pericytes, while vWF is a well-known angiogenic molecule [32,33]. Upon LPS treatment, the expression of ANGPT-1, vWF, and α-SMA decreased. In accordance with TJ proteins, these angiogenic proteins increased in response to BZA treatment (Figure 3). Taken together, the results demonstrate that BZA may contribute to the preservation of the BSCB integrity by inhibiting degradation of the TJ proteins and stabilizing vascularity.

### 2.4. BZA Attenuates Caspase-3-Induced Apoptotic Activity after SCI in Rats

Compared to the sham group, the injury-only group showed increased number of terminal deoxynucleotidyl transferase dUTP nick end labeling (TUNEL)-positive cells in the lesion epicenter at 1 day post-injury (dpi). However, BZA treatment significantly reduced the TUNEL-positive cell counts compared to the injury-only group (Figure 4A,B). To directly examine the activity of the apoptosis executioner caspase-3, western blots were performed. The results showed higher expression of activated caspase-3 in the injury-only group (Figure 4C,D). Conversely, BZA attenuated SCI-induced caspase expression. Taken together, marked differences in TUNEL-positive cells and caspase-3 activity across the experimental groups demonstrate that BZA has neuroprotective role against apoptosis following acute SCI.

### 2.5. BZA Downregulates the Phosphorylation of ERK and p38 Pathways Induced by SCI

Alterations in the activation of MAPKs were investigated in the rat spinal cord tissues at 48 h post-injury by Western blot analysis. The results indicated that BZA significantly attenuated the phosphorylation of ERK and p38 MAPKs (Figure 5). MAPKs are known to be upregulated in response to environmental stresses such as inflammatory stimuli and oxidative stress, playing a pivotal role in mediating SCI progression. Among MAPKs, the p38 pathway is considered to be crucial in the apoptosis network and in mobilizing major SCI-mediated proinflammatory cytokines such as IL-1β, TNF-α, and IL-6 [34,35].

Although mechanisms have not yet been fully elucidated, the ERK1/2 pathway is also known to participate in multiple secondary injury events, such as glutamate excitotoxicity, inflammation, apoptosis, and pain hypersensitivity [36,37]. Moreover, it has been reported that inhibition of ERK pathway promotes OL generation and recovery of demyelinating diseases and that acute strong induction of ERK in adulthood induces demyelination [23,38]. Furthermore, BZA has been shown to improve neurologic deficits in a TBI model and to reduce ischemic lesions in a stroke model by suppressing these MAPKs. Herein, we hypothesized that BZA would provoke similar outcomes in the SCI setting. Hence, as shown in Figure 5, phosphorylation of ERK and p38 significantly increased at 48 h after SCI, and this elevated expression was remarkably diminished upon BZA treatment.

We further investigated the expression of IL-6 by Western blots. IL-6 was barely expressed in the sham group, yet it became significantly abundant after SCI. Surprisingly, the increased expression of IL-6 was suppressed by BZA (Figure 5B,E).

### 2.6. BZA Enhances OPC Differentiation and Remyelination after SCI in Rats

We next assessed remyelination at dpi 20 on the basis of immunofluorescent expression of OPCs, OLs, and myelin markers within the lesion area. Much of the attention on remyelination in the CNS has focused on OPCs and their three phases of response to injury: activation, recruitment, and differentiation [39,40]. Following injury, OPCs first proliferate and migrate to the lesion site, and finally differentiate into myelin-producing OLs. In studies regarding myelination, OPCs and OLs have been variably defined by a multitude of markers as their expression levels illustrate the progression of cell fate specification [39]. Among the commonly used markers for different stages of maturation, oligodendrocyte transcription factor-2 (Olig2), neuron-glial antigen 2 (NG2), oligodendrocyte marker (O4), 2′, 3′-cyclic nucleotide 3′-phosphodiesterase (CNPase), and myelin *oligodendrocyte* glycoprotein (MOG) were evaluated in this experiment. Olig2 is known to be continuously expressed in the differentiation process from neural progenitor cells to mature OLs. NG2, O4, and CNPase represent a transitional stage along the oligodendroglial lineage, which includes a blend of OPCs and oligodendrocytes, while MOG represents mature OLs [41,42]. The presence of myelination was further determined by the LFB staining.

Representative immunofluorescence staining (Figure 6A) and semi-quantification (Figure 6B–F) showed significantly increased expression of Olig2, O4, MOG and increased expression of NG2 and CNPase with no statistical significance in the BZA-treated group. To further validate the effect of BZA on remyelination, the percentage of spared white matter area in the lesion center was measured using LFB staining at dpi 1 and dpi 20 (Figure 6G–J). The percentage of myelinated areas was significantly greater in the BZA group than in the injury-only group. Hence, BZA-treated animals showed increased number of myelin fibers in the lesion center, demonstrating that BZA promotes remyelination after SCI.

### 2.7. BZA Reduces Lesion Volume and Improves Functional Recovery following SCI

The Basso, Beattie, and Bresnahan (BBB) locomotor scale scores were evaluated to assess improvements in motor function at days 1, 3, 7, 10, 14, and 21. The sham group (*n* = 3) scored 21 throughout the experiment. After SCI, complete paralysis was observed in the hindlimbs, with the lowest BBB scores at day 1 in both vehicle (1.29 ± 0.29) and BZA-treated groups (0.75 ± 0.25). Both the injury-only and BZA-treated groups showed gradual recovery in motor function, yet the extent of improvement differed. On days 7, 10, 14, and 21, the vehicle group scored 5.70 ± 0.70, 8.44 ± 0.44, 11.07 ± 1.07, and 14.13 ± 0.88, while the BZA-treated group scored 8.57 ± 0.57, 11.57 ± 1.57, 13.13 ± 1.13, and 17.19 ± 1.81, respectively (Figure 7A). The results indicate that BZA promotes rapid and greater functional recovery of motor function.

At 21 days after SCI, all the spinal cord tissues were harvested. The injured spinal cord surface of the BZA-treated group showed less shrinkage and less loss of tissue volume compared with the injury-only group (Figure 7B). In addition, hematoxylin and eosin (H&E) staining of the injured spinal cords at dpi 1 (Figure 7C) and dpi 20 (Figure 7D) revealed a significant decrease in the percent lesion area with BZA treatment. In the injury-only group, the percent lesion area at dpi 1 and dpi 20 were 17.529 ± 2.554% and 15.726 ± 3.471%, respectively, and 13.447 ± 1.657% and 6.953 ± 1.789% in the BZA-treated group, respectively (Figure 7E). Hence, there was no significant difference in the lesion area between dpi 1 and dpi 20 in the injury group (Figure 7E). In contrast, the BZA-treated group showed a notable decrease in the lesion area from dpi 1 to dpi 20, indicating greater recovery. Moreover, there was no difference in the lesion area between the injury-only group and the BZA group immediately after surgery, but gradual improvement was observed with BZA treatment.

## 3. Discussion

In this study, we assessed the therapeutic efficacy of BZA in the field of SCI for the first time. Thus, far, BZA has shown multiple neuroprotective effects, such as reducing inflammation, improving perfusion, and counteracting apoptosis in stroke and TBI models, mainly by inhibiting MAPK/ERK pathways and modulating ERα and ERβ [19,20,21,43]. Moreover, BZA has been described as an effective remyelinating agent that promotes OL differentiation and proliferation independently of ERs [18]. However, the precise mechanisms underlying these neuroprotective roles have not yet been fully elucidated. Here, we investigated the role of BZA as a neuroprotective agent in SCI both in vitro and in a compressive SCI animal model. In close agreement with previous findings, we demonstrated that BZA suppresses MAPK (p38 and ERK1/2) pathways immediately after SCI, thereby preventing extensive neuronal and OL apoptosis and mitigating proinflammatory responses in the injury penumbra. We also found that BZA minimizes the BSCB disruption in vitro. Furthermore, we confirmed that BZA enhances the proliferation and differentiation of OPCs after the injury, contributing to active remyelination and functional recovery.

Since apoptosis and inflammation are critical contributors to injury exacerbation, early and effective neuroprotection is important. The anti-inflammatory effects of BZA have been previously reported in different clinical settings other than SCI. For instance, BZA induced significant reductions in TBI-induced proinflammatory cytokines (IL-1β, IL-6, COX-2, and TNF-α) [21]. In addition, BZA effectively downregulated IL-6 via the signal transducer and activator of transcription 3 (STAT3) pathway in an atherosclerosis model [44]. Interestingly, ER-β agonists have shown to inhibit inflammation in both in vivo and in vitro models of inflammatory bowel disease via downregulation of ERK, JAK2 and STAT, suggesting the general role of estrogen in inflammatory process [45]. In agreement with these findings, we found that BZA promoted the resolution of inflammation by attenuating NO and LPS-activated pro-inflammatory cytokines (TNF-α, CCL-2, IL-6) while upregulating IL-10, an anti-inflammatory cytokine, in vitro. In the animal model of SCI, BZA-treated rats exhibited significantly reduced expression of IL-6 compared to the injury-only group. In fact, IL-6 is a key cytokine that triggers and exacerbates the inflammatory response by enhancing the expression of other proinflammatory cytokines such as TNF-α and IL-1β [46]. Similarly, TNF-α is known to dampen functional recovery after SCI by increasing NO production, elevating neutrophil infiltration and inflammation, inducing edema, and initiating apoptosis of neurons and OLs [47,48,49]. Many studies have reported IL-10 to be one of the key cytokines that counteract the damage driven by excessive inflammation, contributing to wound healing and tissue repair in the SCI environment [50,51]. By modulating these major inflammatory cascades, BZA can potentially contribute to stabilization of the microenvironment of post-SCI sites. Furthermore, we observed anti-apoptotic effects of BZA in vivo. It is well known that apoptosis, as demonstrated by nuclear DNA fragmentation and caspase activation, is a prominent feature in the SCI. Apoptosis of neurons and OLs contributes to continuing cellular destruction in the injured spinal cord and possibly long-term neurological deficits [52]. In the present study, BZA successfully attenuated caspase-3 activity, leading to a lower number of apoptotic bodies in the lesion site than in vehicle animals. Correspondingly, previous reports have found that BZA induced neuroprotection in transient focal cerebral ischemia via inhibition of caspase-3-mediated apoptosis [19,20]. These data provide novel evidence that BZA exerts neuroprotective effects by mitigating the inflammatory response, apoptosis, and preventing excessive tissue damage after SCI.

In the present study, we also explored the effect of BZA on the expressions of junctional complexes (ZO-1 and occludin) in hCMEC/D3 cells. Since increased vascular permeability and BSCB breakdown contribute to the propagation of the acute inflammatory response after SCI by allowing inflammatory cells into the injury penumbra, maintenance of BSCB integrity and permeability is pivotal [17]. Our results demonstrated that BZA reversed the effects of LPS on both ZO-1 and occludin, suggesting a potential role of BZA in restoration of BSCB function after SCI. It is also in line with a previous finding that BZA treatment alleviated the TBI-induced loss of junction proteins in the cortex of animals [21]. We also found that BZA treatment increased the expression of vascular and angiogenic molecules (ANGPT-1, vWF, and α-SMA), further contributing to the stabilization of BSCB. However, additional evidence from in vivo experiments is required to determine physiological effects of BZA on BSCB disruption, such as edema or recruitment of inflammatory cells to the lesion site. Furthermore, it has recently been found that Baicalin, a natural ingredient attenuates BSCB permeability through PI3K/Akt signaling after spinal cord injury in rat models. Likewise, further experiments are required to fully elucidate the molecular mechanisms of BZA on BSCB disruption [53].

To examine the mechanisms of neuroprotective effects of BZA in SCI, we evaluated the changes in MAPKs, with a focus on ERK1/2 and p38. MAPK pathways have been widely investigated and are known to play leading roles in cell differentiation, growth, apoptosis, and many other phenomena in various pathophysiological events in the CNS [54,55]. Previous findings have demonstrated that SCI activates ERK1/2 and p38 MAPK in microglia/macrophages, and inhibitors of these MAPKs reduce inflammation and partially prevent apoptosis in the early stages of the injury cascades [34]. While p38 is mainly involved in neuronal apoptosis and mobilizing major SCI-related proinflammatory cytokines, ERK1/2 is implicated in multiple aspects of CNS injury pathophysiology. It has been described that ERK overactivity in microglia imposes a detrimental effect on adjacent OLs by inducing proinflammatory mediators such as IL-1β, IL-6, and TNF-α, leading to demyelination [56]. Therefore, inhibition of the ERK pathway has been shown to promote OL generation and recovery from demyelinating diseases [23]. In ischemic stroke studies, BZA reduced the infarct volume by strongly inhibiting the ischemia-induced phosphorylation of ERK1/2 and modulating PI3/AKT in both female and male rats [20,43]. Prevailing evidence has highlighted the importance of modulating p38 and ERK 1/2 to mitigate secondary injury responses and induce remyelination after SCI. In fact, our results correspond well to these findings as BZA treatment significantly suppressed the elevation of SCI-induced phosphorylation of both p38 and ERK1/2 in rats. Interestingly, BZA-treated rats also expressed higher levels of OPC and OL markers (Olig2, NG2, O4, CNPase, and MOG) than the injury-only group at dpi 20, as reported in other models of CNS injuries [18]. To our knowledge, Olig2 is present throughout the oligodendroglial lineage and its gain of function in OPC can promote remyelination [57]. O4 and CNPase are known to represent pre-OLs, while MOG is a surface marker of myelin proteins [39,57]. NG2 commonly serves as reservoir for new OLs, but at the same time, NG2 cells can also differentiate into astrocytes at the injury site and participate in glial scar formation [58]. In our study, the statistical analysis indicated significant differences between the injury-only and BZA-treated groups in the expression of Olig2, O4 and MOG, but not in NG2 and CNPase expression. This may be largely due to the small sample size; hence, further experiments with a larger number of animals would enhance reproducibility. Another consideration is that the stages of OL development are less a sequence of discrete subtypes, and there is not an established point at which a cell is definitively labeled [39]. Therefore, it is difficult to determine which markers best identify OPCs and OLs at this specific time window (dpi 20) and accurately measure the extent of cell accumulation after SCI. In the case of NG2, its cellular roles in pathological conditions remain largely elusive; hence, NG2 may not directly reflect the remyelinating effect of BZA in this experiment [59]. Nonetheless, elevated expression of Olig2, O4 and CNPase demonstrates successful enhancement of subacute remyelination after SCI.

Since this is the first study to examine the neuroprotective effects of BZA in SCI, it is challenging to explain the direct link between these pathways and the described beneficial outcomes. Although not definitive proof, these results indicate the likely possibility that BZA exerts anti-inflammatory and anti-apoptotic effects via downregulation of p38 phosphorylation. Simultaneously, BZA enhances remyelination by promoting OPC proliferation and differentiation in the lesion site via suppression of ERK1/2 pathways (Figure 8A). More importantly, the BBB scores of the sham, injury, and BZA groups demonstrated that daily BZA treatment leads to a faster and greater extent of locomotor function improvement. The findings of this study collectively indicate clearly that BZA is a potential therapeutic candidate for SCI as it exerts multiple neuroprotective roles along the injury cascades.

However, further investigations are needed to corroborate our findings. First of all, it is fundamental to illustrate the complete mechanisms and crosstalk between the described neuroprotective effects of BZA. The first step would be to analyze the involvement of ERs, as they are the main modulatory targets of SERMs. One study investigated whether BZA had an additive effect on OPC differentiation in combination with a previously validated pro-remyelinating compound known to operate through a distinct pathway, to provide insight into the divergent mechanism of BZA [18]. Likewise, searching for the candidate targets through which BZA exerts its effects would deepen our understanding of its role in SCI. Secondly, in order to observe how the remyelinating effect of BZA leads to functional improvements, investigations of axonal regeneration and structure, as well as conduction of electrical impulses, will be valuable. Lastly, further examinations of other major inflammatory pathways such as nuclear factor *kappa B* (NF-κ*B*) and c-Jun N-terminal kinase (JNK), and associated cytokines both in vitro and in vivo will provide better insights into mechanisms behind the inflammatory modulations of BZA.

## 4. Materials and Methods

### 4.1. Cell Cultures, Reagents, Drug and Treatments

A rat adrenal pheochromocytoma cell line PC12 (American Type Culture Collection) and macrophage cell line, RAW 264.7 (Korean Cell Lines Bank, Seoul, Korea) were used to investigate the effect of BZA on cell viability and inflammation. A human cerebral microvascular endothelial cell line (hCMEC/D3, EMD Millipore Corporation, Temecula, CA, USA) was used to study BZA’s impact on angiogenesis, BSCB permeability, and anti-inflammation upon induction of inflammation.

PC12 cells were cultured in RPMI-1640 medium (Thermo Fisher Scientific, Waltham, MA, USA) supplemented with 10% horse serum, 5% fetal bovine serum (FBS) and 1% penicillin/streptomycin. RAW 264.7 macrophage cells were cultured in Dulbecco’s modified Eagle’s medium (DMEM) with 10% FBS and 1% penicillin/streptomycin. hCMEC/D3 cells were cultured in a fully supplemented endothelial growth medium (PromoCell, Heidelberg, Germany). All the cell lines were incubated at 37 °C in an atmosphere of 5% CO_2_.

LPS (Sigma Aldrich, St. Louis, MO, USA), prepared at a concentration of 5 μg/mL was used to induce inflammation in the cells. BZA, an indole-based ER ligand (trade name: Viviant^®^), was donated by CHA Bundang Medical Center and dissolved in 10% DMSO (1 mg/mL stock). The chemical structure of BZA is presented in Figure 8B. For in vitro studies, BZA was treated at concentrations of 10 μM and 20 μM to assess dose-dependent effects. In the in vitro experiments, LPS was treated for 12 h in the LPS-only group; the LPS + BZA groups received pre-treatment with BZA for 12 h, followed by LPS treatment for another 12 h. In animal studies, a 1 mg/kg intraperitoneal (i.p.) bolus was given preoperatively and a 3 mg/kg i.p. bolus was given postoperatively for a maximum of 7 days depending on various experimental groups.

### 4.2. CCK-8 Cell Viability Assay

Cell viability was measured with the CCK-8 kit (Dojindo Laboratories, Kumamoto, Japan). In 96-well plates, 100 μL of PC12 cells (1 × 10^4^) were suspended and incubated for 24 h. Then the cells were treated according to four different experimental groups: the control group, LPS group, and LPS with BZA 10 μM group, and the LPS with BZA 20 μM group. In the treatment groups, the cells were pre-treated with BZA for 12 h and LPS was added for another 12 h for total of 24 h of incubation. After that, 10 μL of CCK-8 solution was added to each well and incubated for 3 h in the 37 °C incubator. Optical density at 450 nm was measured with a microplate reader.

### 4.3. Nitric Oxide (NO) Assay

The inhibitory effects of BZA on NO production were evaluated by measuring nitrate (NO_3_) and nitrite (NO_2_) levels using the NO Plus Detection kitbased on diazotization reaction (iNtRON Biotechnology, Seongnam, Korea). Briefly, RAW264.7 cells were cultured in a 6-well plate (cell density: 0.3 × 10^6^ cells/well) for 24 h and the cells were stimulated with LPS (100 ng/mL) in the presence or absence of BZA (10 μM and 20 μM). After further incubation for 24 h, the cell supernatants were collected from each well and centrifuged to remove cell debris. The assay was carried out following the manufacturer’s protocol. The absorbance was measured at 520 nm and 560 nm. The nitrite concentration was calculated by the standard curve of sodium nitrite.

### 4.4. RNA Isolation and qRT-PCR Analysis

The anti-inflammatory effects of BZA were determined by measuring the mRNA expression of pro-inflammatory and anti-inflammatory cytokines via RT-PCR. Raw 264.7 cells were first prepared on a 6-well culture plate and treated with LPS in the presence or absence of BZA according to the experimental groups. After a 24-h treatment period, total RNA was extracted using TRIzol reagent (Invitrogen, Waltham, MA, USA), and 1 μg of RNA was used to reverse-transcribe cDNA with a cDNA synthesis kit (TAKARA, Shiga, Japan). qPCR was performed using SYBR Green Master Mix kit (Thermo Fisher Scientific, Waltham, MA, USA, #4367659), and amplification was performed by ABI StepOne Real-Time PCR System (Applied Biosystems, Warrington, UK). The relative expression of mRNA was normalized to 18s levels and analyzed by the ΔΔCT method. The primers were synthesized by Bioneer corporation (Bioneer, Daejeon, Korea). A list of primers is given in Table 1.

### 4.5. Immunocytochemistry (In Vitro)

Immunocytochemistry was performed to evaluate the effects of BZA on BSCB permeability and angiogenesis. hCMEC/D3 cells were cultured on a coverslip in 24-well plate for 24 h and then fixed in a 4% paraformaldehyde. The cells were permeabilized using 0.5% Triton-X reagent and blocked with 5% BSA and 0.5% Tween-20 in PBS at room temperature for 1 h. Then, cells were incubated with primary antibodies overnight at 4 °C: zonula occludens-1 (ZO-1, 1:100, Invitrogen, Waltham, MA, USA, #40-2300) and occludin (1:100, Invitrogen, Waltham, MA, USA #33-1500), angiopoietin-1 (ANGPT-1, 1:200, Abcam, Cambridge, UK, ab133425), von Willebrand factor (vWF, 1:50, Millipore, Burlington, MA, USA, AB7356), and alpha smooth muscle actin (α-SMA, 1:200, Abcam, Cambridge, UK, ab21027). After that, the cells were incubated with secondary antibodies at a 1:500 ratio at room temperature for 1 h; donkey anti-rabbit Alexa Fluor^®^647 (Invitrogen, Waltham, MA, USA, A31573) and goat anti-mouse Alexa Fluor^®^ 488 (Abcam, Cambridge, UK, ab150117). Lastly, the cells were mounted after 5 min of incubation in DAPI (1:500, Invitrogen, Waltham, MA, USA, D1356) and were analyzed using a fluorescence microscope (Zeiss 880, Carl Zeiss, Oberkochen, Germany) and ImageJ software (National Institutes of Health, Bethesda, MD, USA).

### 4.6. Animals and Surgical Procedures

In total, 45 female, 8-week-old Sprague Dawley (SD) rats weighing 180–200 g were purchased from Koatech (Koatech, Pyeongtaek, Korea). The animals were housed in a facility at 55–65% humidity with a controlled temperature at 24 ± 3 °C. The rats were acclimatized to the housing environment for at least 1 week prior to the experiment. All experimental procedures were performed according to the Institutional Animal Care and Use Committee (IACUC) of CHA University (IACUC200199) and principles of laboratory animal care [60].

For the surgical procedure to create a compression spinal cord injury model, the rats were first anesthetized with a mixture of Zoletil^®^ (50 mg/kg, i.p., Virbac Laboratories, Virbc, France) and Rompun^®^ (10 mg/kg, i.p., Bayer, Seoul, Korea). Laminectomy was performed by exposing the T10 spinal cord, and then the dorsal surface of the spinal cord was subjected to a moderate, static-weight compression injury for 5 min using a 35-g metal impounder (Figure 8C). Meanwhile, the T9 and T11 spinous processes were clamped stabilized by Allis clamps as previously described [61,62,63,64]. The surgical site was closed by suturing the muscle, fascia, and skin. Lastly, povidone-iodine was applied at the operation site and a prophylactic antibiotic (cefazolin, CKD Pharmaceuticals, Seoul, Korea) was injected subcutaneously to prevent infection. The rats were also given an analgesic (ketoprofen, SCD Pharmaceuticals, Seoul, Korea) to minimize surgical stress and 5 mL of 0.9% sterile saline was injected subcutaneously to keep them hydrated. Micturition was assisted by manually pressing the bladder twice daily until recovery. The rats were kept on a thermostatically regulated heating pad to maintain body temperature during the whole process.

The three different treatment groups were as follows: (1) sham: rats only underwent laminectomy, conserving the spinal cord. Then they received the same amount of vehicle (10% DMSO) injection intraperitoneally every 24 h, since BZA was dissolved in 10% DMSO (*n* = 11); (2) SCI alone (vehicle): rats underwent the compression injury procedure and received the same amount of 10% DMSO every 24 h (*n* = 15); and (3) SCI + BZA: rats received an initial bolus of 1 mg/kg BZA 1 h prior to the injury procedure plus 3 mg/kg BZA postoperatively every 24 h (*n* = 14). The animals were sacrificed at three different time points: dpi 1 (sham; *n* = 3, injury; *n* = 6, BZA; *n* = 5), 2 (*n* = 3/per group) and 20 (sham; *n* = 5, injury; *n* = 6, BZA; *n* = 6) (Figure 8C). Hence, BZA or DMSO was administered daily until either sacrifice or up to 7 days depending on the experimental groups. The drug concentration and time window were carefully selected according to the previous literature [19,21,43,65].

### 4.7. Behavioral Assessment-Hindlimb Locomotor Score

Hindlimb motor function was assessed using the open-field BBB locomotor test on dpi 1, 3, 7, 10, 14, 20 (sham *n* = 3, injury *n* = 6, BZA *n* = 5). The BBB scale ranges from 0 to 21, where 0 represents no hindlimb movement and 21 indicates complete recovery [66]. The animals were observed for 1–2 min on an open-field surface and the scores were rated by two experienced investigators who were blind to the experimental conditions. The scores shown were the average data of two investigators.

### 4.8. Western Blot Analysis

Spinal cord tissues were obtained at dpi 2 after sacrificing rats in a CO_2_ chamber. The total and nuclear proteins were extracted by homogenizing the tissues using an Handy tissue homogenizer (Biofact, Dageon, Korea, CDM-1540) lin a lysis buffer (PRO-PREP^TM^, iNtRON Biotechnology, Seongnam, Korea). Protein concentration was determined by BCA protein assay kit (Pierce, Rockford, IL, USA). An aliquot of 40 μg protein from each sample was separated by 10% SDS-PAGE electrophoresis and the proteins were transferred to polyvinylidene fluoride (PVDF) membranes (Bio-Rad, Hercules, CA, USA, #162-0177). The membranes were incubated in either 5% non-fat skim milk in Tris-buffered saline solution with Tween (TBST) or 5% BSA depending on the antibodies, to block nonspecific binding. Then the membranes were immunoblotted with primary antibodies at 4 °C overnight: ERK (1:1000, Santa Cruz Biotechnology, Dallas, TX, USA, #271269), p-ERK (1:1000, Santa Cruz Biotechnology, Dallas, TX, USA, #7383), p38 (1:1000, Cell Signaling, Danvers, MA, USA, #9212), p-p38 (1:1000, Cell Signaling, Danvers, MA, USA, #4511), caspase-3 (1:500, Abcam, Cambridge, UK, #ab13847), IL-6(1:1000, Abcam, Cambridge, UK, #ab9324), and β-actin (1:1000, Novus Biologicals, Centennial, CO, USA, G043). Subsequently, the membranes were incubated with the corresponding horseradish peroxidase-conjugated secondary antibodies for 1 h: goat anti-mouse IgG antibody (1:2000, Santa Cruz Biotechnology, Dallas, TX, USA, #sc-2005) and mouse anti-rabbit IgG antibody (1:2000, Santa Cruz Biotechnology, Dallas, TX, USA, #sc-2004). After the reaction, membranes were treated with enhanced chemiluminescence reagents (ECL, GE Healthcare, Chicago, IL, USA) and detected using the LAS 4000 biomolecular imager (GE Healthcare, Chicago, IL, USA).

### 4.9. Tissue Processing

All the slide samples used for the TUNEL assay, H&E staining, Luxol fast blue (LFB) staining, and immunofluorescence experiments were made with paraffin blocks. Spinal cord samples were collected by perfusion and centered on the lesion site T9 after anesthetizing the rats with Zoletil. Perfusion was performed using a pump with saline and 4% paraformaldehyde (4% PFA, Biosesang, Seongnam, Korea) through heart. The samples were then fixed in 4% PFA for a day, followed by dehydration with graded ethanol solutions. The process was completed by immersing the samples in 100% xylene and paraffin. Paraffin blocks made in the longitudinal direction were sectioned at 5 μm by a microtome (Leica, Wetzlar, Germany, #RM2255).

### 4.10. TUNEL Assay

To evaluate the anti-apoptotic effect of BZA, a TUNEL assay was performed using the TUNEL assay kit (TACS^®^2 TdT-DAB in Situ Apoptosis Detection kit, Trevigen, Gaithersburg, MD, USA, #4810-30-K). Spinal cords (total length of 3 cm, 1.5 cm caudal and rostral to the epicenter, respectively) were harvested at dpi 1 and preserved in 4% PFA for 24 h. Tissues were processed and embedded in paraffin. Histologic sections (5 μm, longitudinal section) were then deparaffinized, dehydrated and incubated with Proteinase K solution for 1 h. Samples were incubated with quenching solution for 10 min, followed by immersion in 1X TdT labeling buffer for 5 min. After that, the labeling reaction mix was added and incubated for 1 h at 37 °C. Slides were incubated with streptavidin-HRP in a 37 °C humidity chamber shortly. Finally, the sections were immersed in 2,2-diaminobenzidine (DAB) solution for 5 min and counterstained with 1% methyl green. The sections were mounted, and the number of apoptotic cells was assessed with an Olympus C-mount camera (U-TVO.63XC, Tokyo, Japan). TUNEL-positive cells were counted in longitudinal sections of the spinal cord at the epicenter.

### 4.11. Luxol Fast Blue (LFB) Staining

To evaluate the extent of myelination, LFB staining was performed on spinal cord tissue samples of dpi 1 and 20 using Luxol Fast Blue Stain Kit (Abcam, Cambridge, UK, #ab150675). The spinal cord tissue sections were deparaffinized in xylene (Samchun chemicals, Seoul, Korea) and dehydrated in graded ethanol solutions and incubated in LFB solution at room temperature for 24 h. The sections were differentiated in the lithium carbonate solution followed by 70% ethyl alcohol and were then counterstained with cresyl violet. Lastly, the sections were dehydrated in 100% alcohol, cleared in xylene, and mounted using mounting solution (Canada balsam, Junsei, Tokyo, Japan, #23255-1210). Scanning was done with OLYMPUS C-mount camera adapter (U-TVO.63XC, Tokyo, Japan). The total area of cavitation and LFB-positive myelinated areas at the injury epicenter (0 mm), as well as 1–2 mm rostral and caudal areas from the epicenter were measured using the Zeiss analysis program (Zen 3.1 Blue edition, Zeiss, Thornwood, NY, USA).

### 4.12. Hematoxylin and Eosin (H&E) Staining

H&E staining was done to analyze the changes in lesion volume. Spinal cord histological sections obtained at dpi 1 and 20 were deparaffinized and dehydrated as described above. Subsequently, the sections were stained with hematoxylin solution, followed by washing in 1% ammonia water (Ammonia solution, Junsei, Tokyo, Japan). The sections were counterstained with eosin solution for 1 min, dehydrated with graded alcohol, cleared in xylene, and mounted using a mounting solution (Canada balsam, Junsei, Tokyo, Japan, #23255-1210). Scanning was performed using the Zeiss analysis program (Zen 3.1 Blue edition, Zeiss, Thornwood, NY, USA).

### 4.13. Immunohistochemistry (In Vivo)

Immunohistochemistry was performed on the spinal cord of rats sacrificed on dpi 20 to assess remyelination. Paraffin blocks were sectioned (5 μm, longitudinal) and subjected to immunofluorescence staining. After deparaffinization, the samples underwent pepsin treatment and blocking (1 h), followed by overnight incubation in primary antibodies. On the next day, the slides were stained with secondary antibodies and 300 mM DAPI (1:500, Invitrogen, Waltham, MA, USA, #D1306), and finally mounted with a fluorescent mounting medium (Dako fluorescence mounting medium, Agilent, Santa clara, CA, USA, #S3023).

The primary antibodies used in this experiment were CNPase (1:50, Sigma Aldrich, St. Louis, USA, #C5922), O4 (1:100, Millipore, #3108670), MOG (1:300, Novus Biologicals, Centennial, CO, USA, #AF2439) and NG-2 (1:500, Abcam, Cambridge, UK, #ab129051). The secondary antibodies were as follows: goat anti-rabbit IgG Alexa fluor 488 (1:300, Invitrogen, Waltham, MA, USA, #A-11034), goat anti-mouse IgG Alexa fluor 488 (1:300, Invitrogen, Waltham, MA, USA, #A-11029), goat anti-mouse IgM Alexa fluor 488 (1:300, Invitrogen, Waltham, MA, USA, #A-21042), and donkey anti-goat IgG Alexa fluor 568 (1:300, Abcam, Cambridge, UK, #ab175704). The immunofluorescence samples were examined using a Digital Slide Scanner (Zeiss Axio Scan. Z1, Carl Zeiss, Oberkochen, Germany). After that, the scanner data were quantified using the Zeiss analysis program (Zen 3.1 Blue edition, Zeiss, Thornwood, NY, USA).

### 4.14. Statistical Analysis

All data are expressed as mean ± standard error of the mean (SEM). The data were compared by one-way analysis of variance (ANOVA), followed by the Bonferroni test. The BBB scoring data were tested for significance using two-way ANOVA, followed by the Bonferroni test and expressed with standard deviations (SDs). H&E staining data were verified using Student’s t-test for comparisons within each group. All data were analyzed with GraphPad Prism (version 5.01, GraphPad Software, La Jolla, CA, USA). A *p-*value < 0.05 was considered to indicate statistical significance. The details and statistical significance are indicated in each figure legend.

## 5. Conclusions

We demonstrated for the first time that BZA has neuroprotective and remyelinating roles in a rat SCI model. BZA promotes inflammatory resolution, remyelination, and functional recovery by modulating p38 and ERK pathways after SCI. The evidence includes downregulation of proinflammatory cytokines, preservation of BSCB via TJ and vasculature proteins, and enhancement of OPC differentiation and OL proliferation. These effects of BZA ultimately led to improvements in locomotor function. Thus, BZA may be a possible therapeutic candidate for SCI.

## Figures and Tables

**Figure 1 ijms-22-11012-f001:**
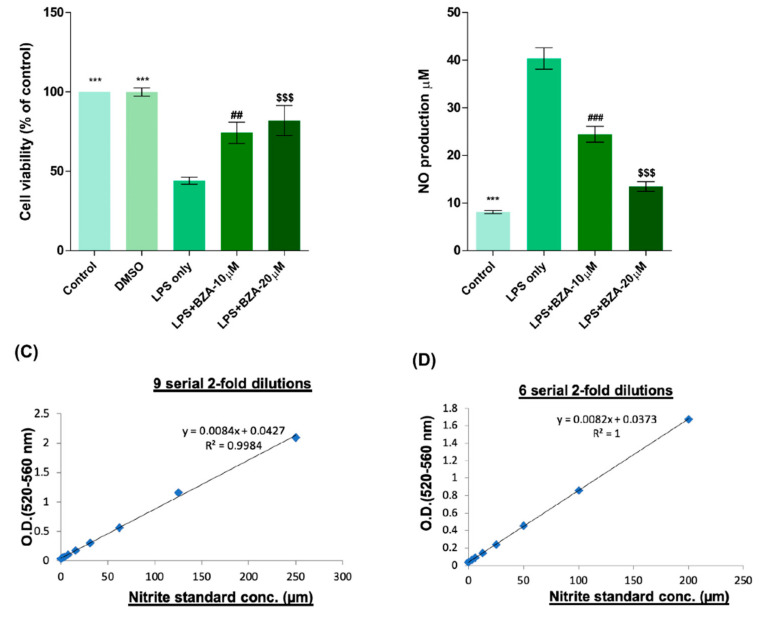
BZA promotes survival of PC12 neural cells and reduces in vitro NO production. Cell viability was significantly higher in LPS + BZA-treated cells compared with LPS-treated cells. NO production significantly decreased in response to 10 μM and 20 μM BZA treatment compared with LPS-treated cells. (**A**) Quantitative bar graph illustrating the percentage of live PC12 neural cells according to treatment group at 24 h following culture. (**B**) Quantitative bar graph illustrating inhibition of NO production in LPS-stimulated RAW264.7 cells treated with BZA. Standard curves to confirm the correlation between NO concentration and absorbance were obtained in two ways: (**C**) 9 serial 2-fold dilutions and (**D**) 6 serial 2-fold dilutions. Data are presented as mean ± SEM (*n* = 3 (**B**–**D**), *n* = 7 (**A**); performed in triplicate). *** *p* < 0.001 (LPS-treated vs. control, DMSO), ## *p* < 0.01, ### *p* < 0.001 (LPS-treated vs. LPS + BZA 10 μM), $$$ *p* < 0.001 (LPS-treated vs. LPS + BZA 20 μM), one-way ANOVA followed by the Bonferroni test.

**Figure 2 ijms-22-11012-f002:**
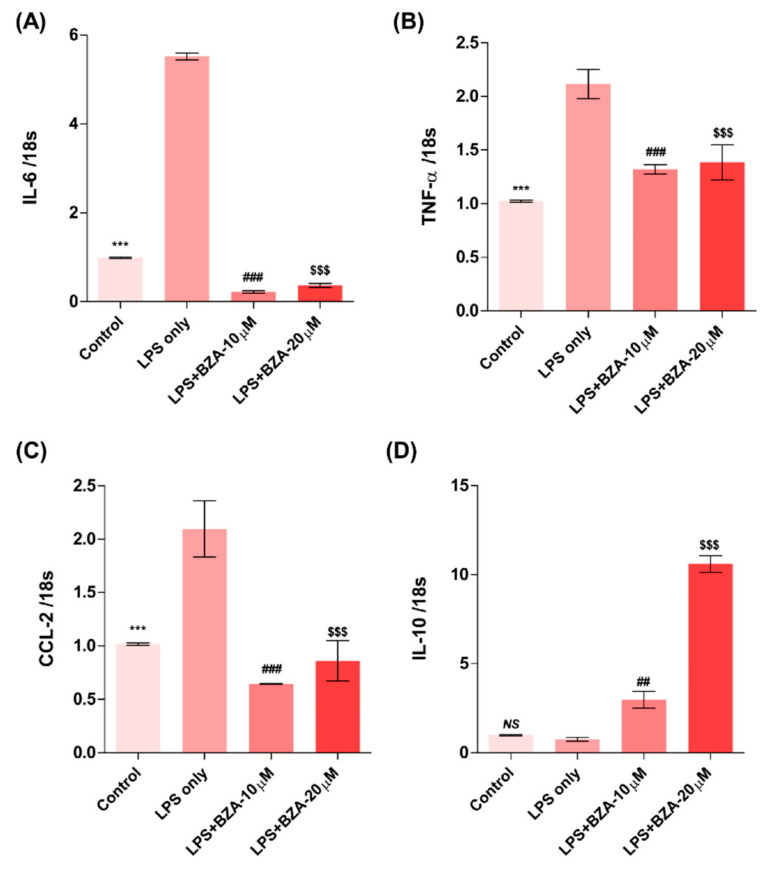
BZA promotes the resolution of inflammation in vitro. Total RNA was extracted from the LPS- and BZA (10 μM and 20 μM)-treated RAW 264.7 cells at 24 h following incubation (*n* = 3, performed in triplicate). The mRNA levels of proinflammatory cytokines including IL-6 (**A**), TNF-α (**B**), and CCL2 (**C**) and an anti-inflammatory cytokine, IL-10 (**D**), were measured. For time, 18 s was used as internal control for qRT-PCR. Data represent mean ± SEM (*n* = 3 (**A**,**C**), *n* = 4 (**D**), *n* = 6 (**B**); performed in triplicate). *** *p*< 0.001 (LPS-treated vs. control, DMSO), ## *p* < 0.01, ### *p* < 0.001 (LPS-treated vs. LPS + BZA 10 μM), $$$ *p* < 0.001 (LPS-treated vs. LPS + BZA 20 μM)*, NS* = Not significant, one-way ANOVA followed by the Bonferroni test.

**Figure 3 ijms-22-11012-f003:**
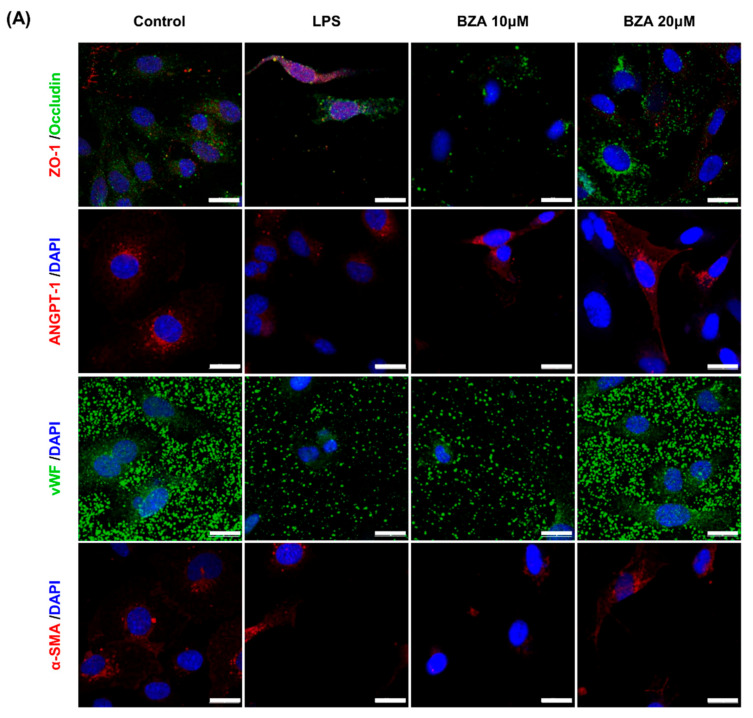
BZA increases angiogenesis and decreases BSCB destruction in vitro. (**A**) Representative immunofluorescent images (scale bar = 20 μm) of tight junction proteins such as ZO-1 and occludin and angiogenesis markers including ANGPT-1, vWF, and α-SMA examined in hCMEC/D3 cells. Quantitative fluorescence intensity for (**B**) ZO-1, (**C**) occludin, (**D**) ANGPT-1, (**E**) vWF, and (**F**) α-SMA. Data represent mean ± SEM (*n* = 3 (**A**,**C**), *n* = 4 (**D**), *n* = 6 (**F**); performed in triplicate). ** *p* < 0.01, *** *p* < 0.001 (LPS-treated vs. control, DMSO), ## *p* < 0.01 (LPS-treated vs. LPS + BZA 10 μM), $ *p* < 0.05, $$ *p* < 0.01, $$$ *p* < 0.001 (LPS-treated vs. LPS + BZA 20 μM)*, NS* = Not significant, one-way ANOVA followed by the Bonferroni test.

**Figure 4 ijms-22-11012-f004:**
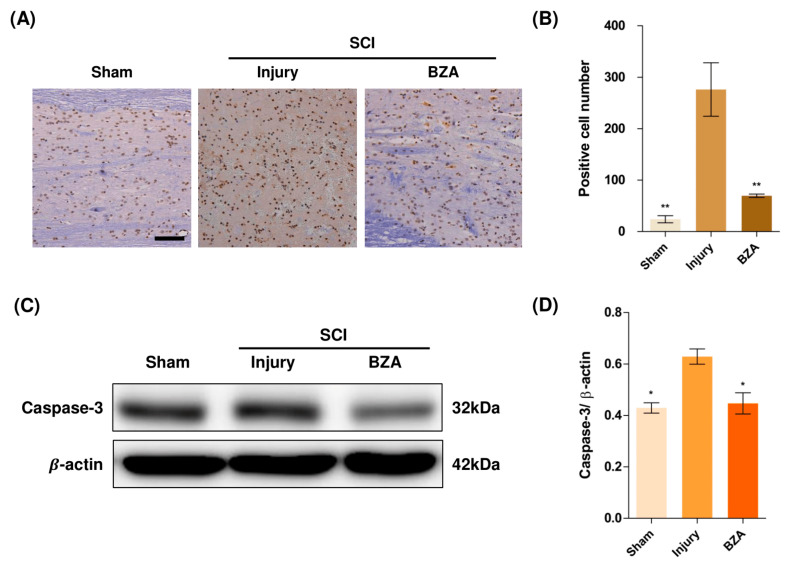
BZA attenuates caspase-3-induced apoptosis activity after SCI and suppresses the inflammatory response. (**A**) Representative images of the TUNEL assay observed in each group of spinal cord samples to examine apoptosis activity. (**B**) Graph result for positive cell number by quantitative analysis of TUNEL assay. To confirm the activity against apoptosis, western blot for caspase 3 was performed. (**C**) Representative images of western blot results for caspase 3 in the sham, injury, and BZA groups. (**D**) The graph of caspase 3 was measured to quantify the western blot results**.** All results of the graph confirmed the significance between the injury group and all other groups. * *p* < 0.05, ** *p* < 0.01, *NS* = Not significant. *N* = 3 (sham), 6 (injury), 5 (BZA) in the TUNEL assay and 3(C–D). TUNEL assay scale bar = 20 μm). Data represent mean ± SEM (in TUNEL, *n* = 3 (sham), 6 (injury), 5(BZA) and *n* = 3; (C,D) performed in triplicates). * *p* < 0.05, ** *p* < 0.01 (injury vs. sham, BZA), *NS* = Not significant, one-way ANOVA followed by the Bonferroni test.

**Figure 5 ijms-22-11012-f005:**
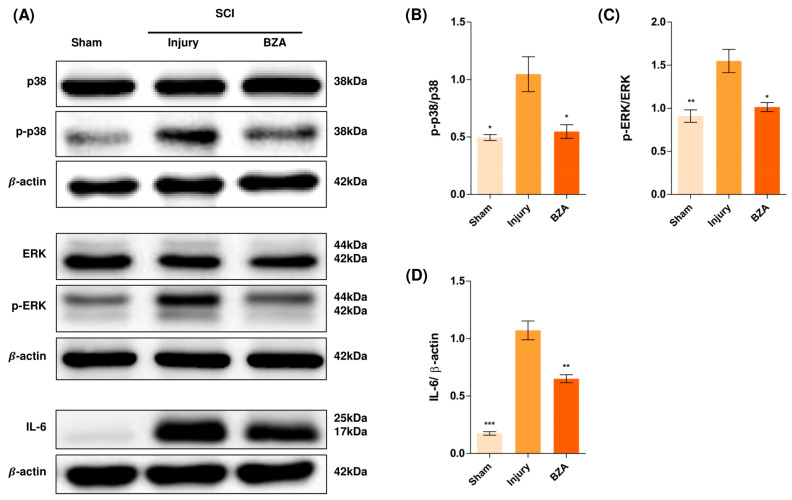
BZA downregulates the phosphorylation of ERK and p38 pathways. BZA decreases phosphorylation of ERK1/2 and p38, resulting in neuroprotection, OPC differentiation, and remyelination. (**A**) Representative images of western blots for p38, p-p38, ERK, p-ERK, IL-6. The graph was quantified as (**B**) p-p38/p38, (**C**) p-ERK/ERK and (**D**) IL-6. Western blot results of the graph confirmed the significance between the injury group and the sham and BZA groups. All data represent mean ± SEM (*n* = 3; performed in triplicate). * *p* < 0.05, ** *p* < 0.01, *** *p* < 0.001 (injury vs. sham, BZA), *NS* = Not significant, one-way ANOVA followed by the Bonferroni test.

**Figure 6 ijms-22-11012-f006:**
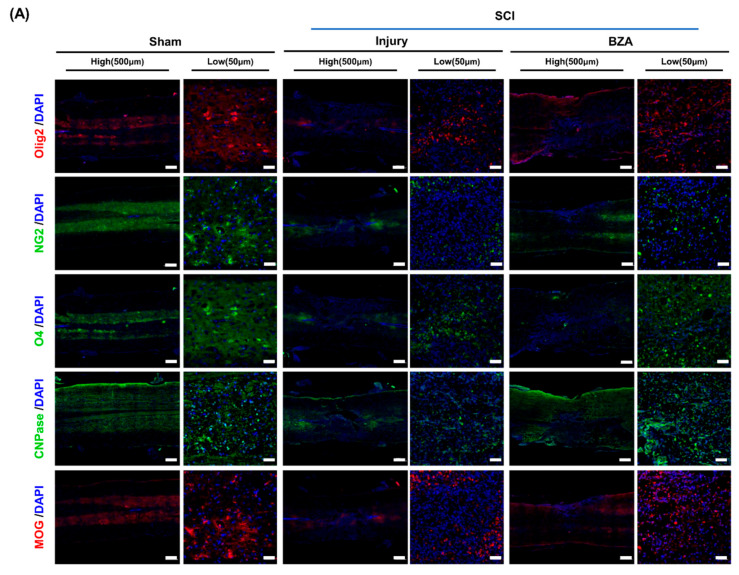
BZA enhances remyelination after SCI. For a quantitative analysis of the formation of oligodendrocytes responsible for myelination, the spinal cord samples at dpi 20 were sectioned and immunostained. (**A**) Representative images of oligodendrocyte differentiation marker Olig2, the mature oligodendrocyte markers NG2, O4, CNPase, and the myelin marker MOG in the sham, injury, and BZA groups. (B–F) The percentage (%) of each antibody area to the total area of the immunofluorescence (IF) quantification graph. (**B**) Olig2, (**C**) NG2, (**D**) O4, © CNPase. (**F**) MOG. (**G**,**H**) Representative images of Luxol fast blue (LFB) staining at dpi 1 and dpi 20 in the sham, injury, and BZA groups. (**I**) Percentage graph of white matter that could be confirmed as myelin compared to the total area of sham, injury, and BZA groups at dpi 20, (**J**) A graph confirming the number of positive cells representing nerve cells at dpi 20. IF results of the graph confirmed the significance between the sham group and the injury and BZA groups. (LFB results = injury vs. sham, BZA). Data represents mean ± SEM (*n* = 3–4 (**B**,**C**), 3 (**D**), 5–6 (**E**), 3–6 (**F**), 4 (**G**–**J**)/group). * *p* < 0.05, ** *p* < 0.01, *** *p* < 0.001 (sham vs. injury, BZA; (**A**–**F**), injury vs. sham, BZA; (**G**–**J**). # *p* < 0.05 (injury vs. BZA; (**A**–**F**), *NS* = Not significant, one-way ANOVA followed by the Bonferroni test.

**Figure 7 ijms-22-11012-f007:**
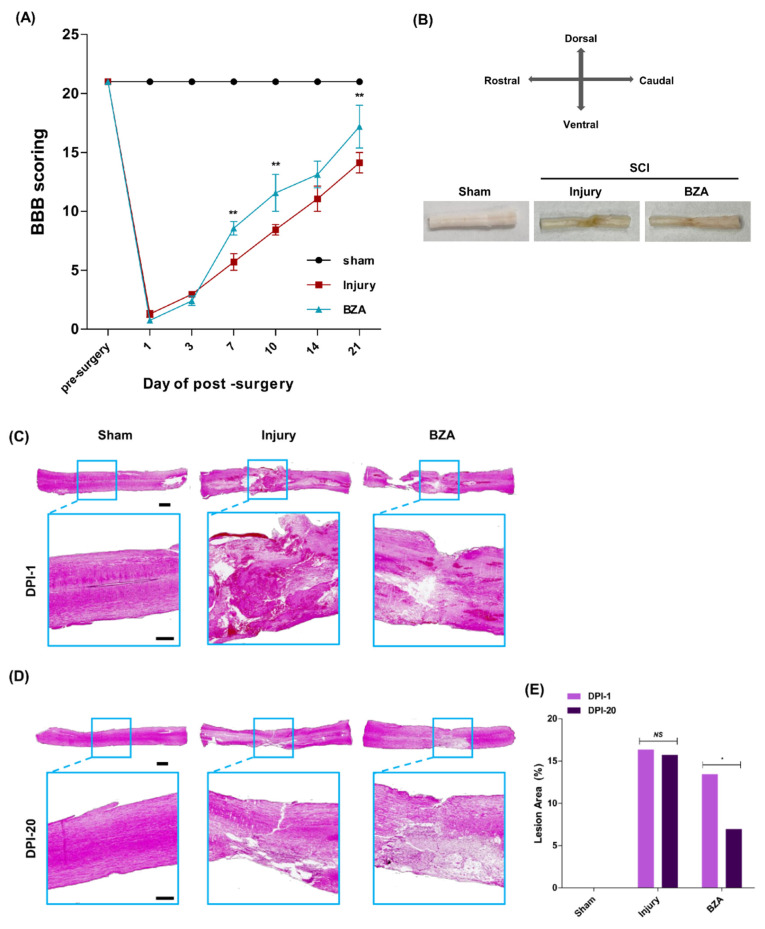
BZA reduces lesion volume and improves functional recovery following SCI in vivo**.** (**A**) Locomotor functional recovery evaluation using BBB scores. The BZA-treated group had significantly higher BBB scores compared with the injury-only group. (**B**) Gross morphology at 21 days after SCI showed better tissue preservation in the BZA-treated group. Representative images of hematoxylin and eosin (H&E) staining at dpi 1 (**C**) and dpi 20 (**D**) and quantitative analys© (**E**) showed reduced tissue necrosis area in the BZA-treated group. Data represent mean ± SEM (*n* = 3(**C**–**E**). Data represent mean ± SD (*n* = 3–6 (**A**); performed by two blinded reviewers with BBB scoring). * *p* < 0.05, ** *p* < 0.01 (dpi 1 vs. dpi 20, same group (**C**), injury vs. BZA (**D**), *NS* = Not significant, Student t test followed by the Mann Whitney test for H&E lesion area percentage and two-way ANOVA followed by the Bonferroni test for BBB scoring.

**Figure 8 ijms-22-11012-f008:**
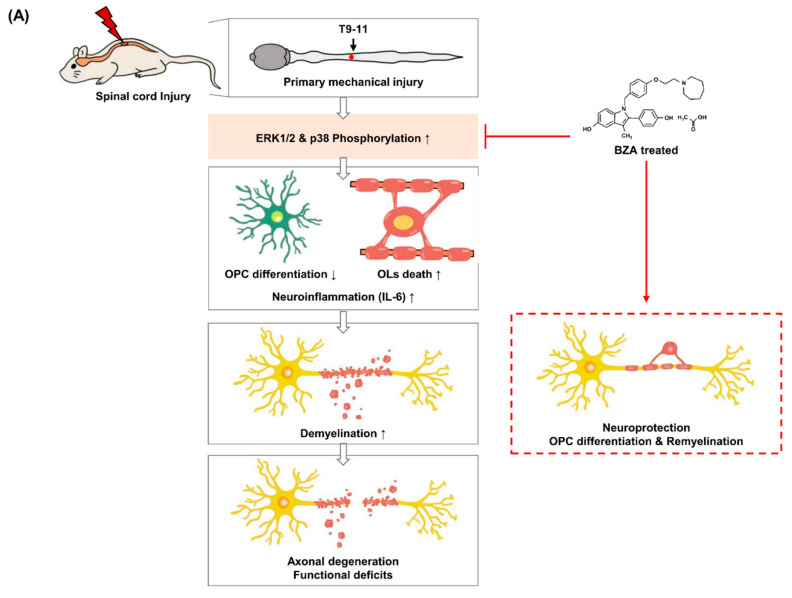
Schematics of the predicted mechanisms of BZA and experimental design. Illustration of spinal cord injury (SCI) modeling and the scheme of the overall experiment. (**A**) Schematic diagram of the effect of BZA after SCI. (**B**) Chemical structure of BZA. (**C**) Creation of moderate static compressive SCI and intraperitoneal (i.p.) injection of treatment materials. (**D**) Schematic overview of the total experimental design.

**Table 1 ijms-22-11012-t001:** Primer sequences for the genes of interest and reference gene.

Primers	Direction	Sequences
TNF-α	Forward	5′-AGCAAACCACCAAGTGGAGGA-3′
	Reverse	5′-GCTGGCACCACTAGTTGGTTGT-3′
CCL-2	Forward	5′-GAAGGAATGGGTCCAGACAT-3′
	Reverse	5′-ACGGGTCAACTTCACATTCA-3′
IL-6	Forward	5′-GCTACCAAACTGGATATAATCAGGA-3′
	Reverse	5′-CCAGGTAGCTATGGTACTCCAGAA-3′
IL-10	Forward	5′-ACTGGCATGAGGATCAGCAG-3′
	Reverse	5′-CTCCTTGATTTCTGGGCCAT-3′
18s	Forward	5′-GCAATTATTCCCCATGAACG-3′
	Reverse	5′-GGCACTTAATCAACGCAAGC-3′

TNF-α, tumor necrosis factor-alpha; CCL-2, C-C motif chemokine ligand-2; IL-6, interleukin-6; IL-10, interleukin-10; 18 s, 18 s rRNA.

## Data Availability

Data available in a publicly accessible repository.

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
