# Peer review of "Bazedoxifene, a Selective Estrogen Receptor Modulator, Promotes Functional Recovery in a Spinal Cord Injury Rat Model"

_ijms, 2021, doi:10.3390/ijms222011012_

Round 1

Reviewer 1 Report

Well written article on the pleiotropic effects of bazedoxifene on functional recovery in a spinal cord injury rat model. Also, nice discussion on the potential mechanisms of action involved in neuroprotection/remyelination. It would be interesting to determine the role of gender on the degree of functional recovery.

The only correction needed is Figure 5 legend where the authors mistakenly wrote that BZA increases phosphorylation of ERK1/2 and p38...

Author Response

We appreciate your thorough and kind review. We have edited Figure 5 legend as you pointed out. Thank you.

[Line 849: BZA decreases phosphorylation]

Reviewer 2 Report

The Abstract of the paper is a honest summary of a well designed and implemented research.

I have no major criticisms or suggestions, but for the GENERAL READERS I strongly suggest to add and discuss a few more REFERENCES, specifically:

Zhao R, Wu X, Bi XY, Yang H, Zhang Q. Baicalin attenuates blood-spinal cord barrier disruption and apoptosis through PI3K/Akt signaling pathway after spinal cord injury. Neural Regen Res. 2022 May;17(5):1080-1087. doi: 10.4103/1673-5374.324857. PMID: 34558536

  Jiang Q, Li W, Zhu X, Yu L, Lu Z, Liu Y, Ma B, Cheng L. Estrogen receptor beta alleviates inflammatory lesions in a rat model of inflammatory bowel disease via down-regulating P2X7R expression in macrophages. Int J Biochem Cell Biol. 2021 Oct;139:106068. doi: 10.1016/j.biocel.2021.106068. Epub 2021 Aug 28. PMID: 34464722  

Edmunds, K.J.; Gargiulo, Imaging Approaches in Functional Assessment of Implantable Myogenic Biomaterials and Engineered Muscle Tissue. Eur. J. Transl. Myol. 2015, 25, 4847. [CrossRef] [PubMed]

Sajer S. Mobility disorders and pain, interrelations that need new research concepts and advanced clinical commitments. Eur J Transl Myol. 2017 Dec 5;27(4):7179. doi: 10.4081/ejtm.2017.7179. eCollection 2017 Dec 5. PMID: 29299226 Free PMC article.

Author Response

Thank you for thorough review and further suggestions. We revised the content of the discussion based on the reference you gave. In the references you provided, we have added content that is particularly relevant to our research. However, we did not quite see how the other 2 references related to our research topic. If you could please explain to us their relevance, we will revise as you instruct. Thank you again

[Line 232-234] Interestingly, ER- agonists have shown to inhibit inflammation in both in vivo and in vitro models of inflammatory bowel disease via downregulation of ERK, JAK2 and STAT, suggesting the general role of estrogen in inflammatory process [45].

[Line 267-270] Furthermore, it has recently been found that Baicalin, a natural ingredient attenuates BSCB permeability through PI3K/Akt signaling after spinal cord injury in rat models. Likewise, further experiments are required to fully elucidate the molecular mechanisms of BZA on BSCB disruption [53].